# Landscape of Genome-Wide DNA Methylation of Colorectal Cancer Metastasis

**DOI:** 10.3390/cancers12092710

**Published:** 2020-09-22

**Authors:** Carmen Ili, Kurt Buchegger, Hannah Demond, Juan Castillo-Fernandez, Gavin Kelsey, Louise Zanella, Michel Abanto, Ismael Riquelme, Jaime López, Tamara Viscarra, Patricia García, Enrique Bellolio, David Saavedra, Priscilla Brebi

**Affiliations:** 1Laboratory of Integrative Biology (LIBi), Centro de Excelencia en Medicina Traslacional (CEMT), Scientific and Technological Bioresource Nucleus (BIOREN), Universidad de La Frontera, Temuco 4810296, Chile; carmen.ili@ufrontera.cl (C.I.); kurt.buchegger@ufrontera.cl (K.B.); hannah.Demond@babraham.ac.uk (H.D.); louise.zanella@ufrontera.cl (L.Z.); jaime.lopez@ufrontera.cl (J.L.); tamara.viscarra@ufrontera.cl (T.V.); 2Departamento Ciencias Básicas, Facultad de Medicina, Universidad de La Frontera, Temuco 4811230, Chile; 3Epigenetics Programme, The Babraham Institute, Babraham Research Campus, Cambridge CB22 3AT, UK; Juan.Castillo-Fernandez@babraham.ac.uk (J.C.-F.); gavin.Kelsey@babraham.ac.uk (G.K.); 4Centre for Trophoblast Research, University of Cambridge, Cambridge CB2 1TN, UK; 5Scientific and Technological Bioresource Nucleus (BIOREN), Universidad de La Frontera, Temuco 4811230, Chile; michel.abanto@ufrontera.cl; 6Instituto de Ciencias Biomédicas, Facultad de Ciencias de la Salud, Universidad Autónoma de Chile, Temuco 4810101, Chile; ismael.riquelme@uautonoma.cl; 7Department of Pathology, Faculty of Medicine, Pontificia Universidad Católica de Chile, Santiago 8330034, Chile; pgarciam@uc.cl; 8Departamento Anatomía Patológica, Facultad de Medicina, Universidad de La Frontera, Temuco 4781180, Chile; enrique.bellolio@ufrontera.cl; 9Departamento de Medicina Interna, Hospital Hernán Henríquez Aravena, Temuco 4781151, Chile; david.saav.p@gmail.com; 10Clínica Alemana de Temuco, Temuco 4810297, Chile

**Keywords:** colorectal cancer, metastasis, DNA methylation, lymph node, genome-wide analysis

## Abstract

**Simple Summary:**

Colorectal cancer is one of the most common neoplasia worldwide. Metastasis in lymph nodes and distant organs indicates poor prognosis; However, the influence of DNA methylation over colorectal metastasis is not well understood. We investigated the genome-wide DNA methylation profile of normal, primary tumour and lymph node metastasis of colon, finding a specific signature of early metastasis, present in primary tumour, that allowed a better understanding of colon cancer spread. In addition, the hypermethylation of FIGN, HTRA3, BDNF, HCN4 and STAC2 could be utilised in primary tumour as biomarkers of colorectal cancer prognosis.

**Abstract:**

Colorectal cancer is a heterogeneous disease caused by both genetic and epigenetics factors. Analysing DNA methylation changes occurring during colorectal cancer progression and metastasis formation is crucial for the identification of novel epigenetic markers of patient prognosis. Genome-wide methylation sequencing of paired samples of colon (normal adjacent, primary tumour and lymph node metastasis) showed global hypomethylation and CpG island (CGI) hypermethylation of primary tumours compared to normal. In metastasis we observed high global and non-CGI regions methylation, but lower CGI methylation, compared to primary tumours. Gene ontology analysis showed shared biological processes between hypermethylated CGIs in metastasis and primary tumours. After complementary analysis with The Cancer Genome Atlas (TCGA) cohort, *FIGN*, *HTRA3*, *BDNF*, *HCN4* and *STAC2* genes were found associated with poor survival. We mapped the methylation landscape of colon normal tissues, primary tumours and lymph node metastasis, being capable of identified methylation changes throughout the genome. Furthermore, we found five genes with potential for methylation biomarkers of poor prognosis in colorectal cancer patients.

## 1. Introduction

Colorectal cancer (CRC) is a heterogeneous disease caused by both genetic and environmental factors [1]. It is the third most commonly diagnosed cancer worldwide (1.85 million cases per year) and is ranked second regarding the global number cancer deaths (880,792 deaths per year) [2]. The incidence varies geographically, with the highest incidence rates found in Australia, New Zealand, Europe, USA and Canada. However, incidence rates are increasing in other regions, including Central and South America [3]. In Chile, CRC incidence has been estimated at 17.0 per 100,000, according to GLOBOCAN data in 2018 [2].

The majority of CRC occurrences are sporadic (70–80%) and related to risk factors such as age and lifestyle without an existent family history or genetic predisposition [4]. Reasons explaining the increased incidence of CRC in many parts of the world include an ageing population and a “Western” lifestyle [5]. In particular, environmental factors, such as high consumption of alcohol, animal fat, processed and red meat combined with a low intake of vitamin D, fibre and fish, smoking and reduced physical activity are thought to increase the risk of CRC [6].

Next to genomic mutations, epigenetic alterations have also been shown to contribute to CRC carcinogenesis in some cases [7,8]. DNA methylation is the most studied epigenetic modification in human cancers. Aberrant DNA methylation found in cancer has been associated with gene inactivation in DNA damage repair and other signalling pathways [9,10]. At a genome-wide level, the methylome of cancer cells has been shown to undergo characteristic changes. Whereas in somatic cells, the majority of the genome is highly methylated, cancer cells undergo global DNA hypomethylation causing chromatin rearrangements and decondensation [11,12]. In contrast, regions with commonly low methylation levels, such as the CpG-rich CpG islands (CGIs), are frequently hypermethylated in cancer cells. These regions often overlap regulatory sites such as gene promoters and enhancers and their hypermethylation can be correlated with transcriptional silencing [13].

DNA methylation has been widely studied in CRC. In fact, the recognition of differences in the methylation patterns of CRC patients has led to the CpG island methylator phenotype (CIMP) classification. Patients classified as CIMP-high (CIMP-H) or CIMP 1 have three or more specific marker loci methylated. This type of cancer has high microsatellite instability (MSI) in patients who do not have germline mutations in mismatch repair (MMR) genes and have a better prognosis [14,15]. When less than three loci are methylated and *KRAS* mutations are present, patients are classified as CIMP-low (CIMP-L) or CIMP 2 in recognition of the lower levels of methylation and poor prognosis [15].

In many tumour types, the presence of distant metastases marks stage IV of carcinogenesis and indicates, almost invariably, incurable disease and relatively short overall survival [16]. In CRC, tumour stage is mainly determined by the tumour’s ability to spread to lymph nodes, the liver or both [17]. Metastasis formation as well as the type of metastasis determine tumour prognosis. It is estimated that 20% of patients with CRC have metastases at the time of diagnosis and that another 20–25% of patients will experience metastases during the course of the disease. Together, this results in a relatively high overall mortality rate of 40–45% [18]. Liver metastasis formation is the decisive and most lethal event during CRC development. It is not altogether clear which mechanisms drive the formation of metastases. While genomic mutations have been well characterised as the main drivers of early carcinogenesis, metastasis formation is not associated with additional mutations. Instead, other pathways have been proposed, although so far not completely satisfactory. Metastasis is associated with dysregulation of signalling pathways and dysfunction of many molecules, but they cannot fully explain metastasis formation [19]. Another mechanism that may drive metastasis formation is epigenetic deregulation, as gene promoter DNA hypermethylation is well known to affect a number of genes in CRC pathogenesis and liver metastasis [20]. Analysing the DNA methylation changes occurring during CRC progression and early metastasis formation is crucial for the identification of novel epigenetic markers that could improve detection, prognosis, and therapy [21].

To improve our understanding of the metastatic process the present study profiled DNA methylation in colorectal cancer and lymph node metastasis, using next generation sequencing. By mapping the methylation landscape of colon normal tissues, primary tumours and lymph node metastasis, we identified methylation changes throughout the genome. We associated regions of CGI hypermethylation with their nearby genes and with patient survival, thereby identifying novel candidate regions to use as early prognostic markers for colon cancer metastasis in the future.

## 2. Results

### 2.1. Clinical and Pathological Characteristics of Patient Cohort

We collected formalin-fixed paraffin-embedded (FFPE) paired tissues from five CRC patients, histologically classified in a TNM stage between IIIB–IIIC. Tissues collected were primary colon tumour, normal adjacent tissue (NAT) and at least one near to tumour lymph node with metastasis (LNM). Patients’ age range was between 39 and 95 years old. Three men and two women were included in the study (Appendix A). Tumours were moderately or poorly differentiated tubular adenocarcinomas, and tumour infiltration was classified as serosa or subserosa (Appendix A). DNA from all samples was extracted and bisulfite sequenced.

### 2.2. DNA Methylation QC Analysis

To analyse DNA methylation changes during CRC progression in cancer patients, methyl-seq libraries were generated. Between ~19 and 83 million reads were obtained for each library in total (Appendix A). Of these, uniquely aligned reads (~21 million per library) were deduplicated, resulting in an average of ~18 million reads per library available for whole-genome analysis (Appendix A). One sample (Patient 3 LNM) was excluded for analysis, based on a minimal threshold of at least 10 million reads to include libraries for data analysis.

To analyse DNA methylation, the genome was binned into consecutive 200 CpG windows and average methylation of each window was calculated. Unbiased clustering using principal component analysis showed the LNM sample of Patient 1 as an outlier, driving the main variation in PC1 (Appendix A). Looking at this sample in more detail showed that it had lower average methylation than all other samples, which was the result of large hypomethylated regions, spread throughout the genome (Appendix A). Retrospective analysis of the histology of the LNM samples also showed a different histology of the LNM sample of Patient 1 compared to the other four patients (Appendix A). This sample was therefore considered an outlier. To assess methylation changes throughout tumour development within patients, only Patients 2, 4 and 5, which showed homogenous methylation values and for which we had good quality data for all three stages (NAT, primary tumour and LNM) were included for subsequent analysis, unless stated otherwise. Global DNA methylation in these three sample pairs showed lower levels for primary tumour samples (mean methylation 73.6%) compared to NAT (76.24%) and LNM (76.33%) samples (Appendix A).

### 2.3. Global Hypomethylation of Non-CGI Regions in Primary Tumour but Not Metastasis Samples

The methylome of somatic cells can be distinguished into two different categories: the CpG-rich, unmethylated CGIs, which often correspond with promoters and enhancers, and the CpG-poor, highly methylated non-CGI regions, which cover gene bodies and intergenic regions. Due to the different functions and methylation characteristics, these two categories were analysed separately. To assess non-CGI regions, consecutive 200 CpG windows not overlapping CGIs were used. Average non-CGI methylation appeared lower in primary colon tumour samples, compared to normal samples, although changes did not reach significance (Figure 1A,B; one-way ANOVA, *p* = 0.1360). Surprisingly, the decrease in methylation observed in primary tumour samples was not observed in LNM samples, which showed similar methylation levels to NAT samples (Figure 1A,C). In line with this, principal component analysis showed separate clustering of primary colon tumour but not LNM samples on PC1 explaining 25% of total variation (Figure 1D). LNM samples were split from normal samples by PC4 which represented 10% of total variation (Figure 1D). However, we cannot exclude the possibility that the metastasis samples were contaminated by infiltrating immune cells from the lymph nodes, resulting in a metastasis methylation pattern that is more similar to non-tumour cells (probably due to low cellularity).

Nevertheless, the underlying tumour signature is still evident in LNM as demonstrated by PC4. When analysing the methylation of the windows driving PC4, a decrease in global methylation in both primary tumour and LNM samples was observed (Appendix A). However, the decrease compared to NAT was more pronounced in the primary tumour than in the LNM samples. Generally, the methylation changes in non-CGI regions of primary tumour samples are relatively small (Figure 1A,B), but when looking at the genome they span large blocks (Figure 1E), which are similar to the outlier LNM sample of Patient 1 (Appendix A). The loss of DNA methylation might be associated with CpG density, as CpGs are underrepresented in regions losing DNA methylation in primary tumour samples, whereas they are overrepresented in regions with stable DNA methylation levels (Appendix A).

Taken together, we observed decreased levels of methylation in non-CGI regions in primary tumour samples compared to normal colon epithelium of the same patients. We found a less pronounced, but similar pattern in LNM, which so far have been poorly described.

### 2.4. Identification of Hypermethylated CGIs in Primary Tumour and Lymph Node Metastasis Samples

The majority of CGIs in the genome have very low methylation. However, a subset of CGIs can be highly methylated in a cell-type specific manner. To analyse DNA methylation of CGIs in our CRC dataset, average methylation was calculated for each CGI.

CGIs were categorised depending on their methylation levels found in NAT, into unmethylated CGIs (<40% methylation; *n* = 7342) and highly methylated CGIs (>80% methylation; *n* = 1342). Methylation of lowly methylated CGIs was increased in primary tumour and LNM samples compared to NAT, whereas highly methylated CGIs appeared more stable in the tumour samples (Figure 2A). Primary tumour samples showed a larger effect than LNM samples. Similar to the finding with non-CGI regions, principal component analysis clustered the primary tumour samples apart from NAT samples on PC1, whereas LNM and NAT were split by PC4 (Appendix A). CGIs were found mostly in gene promoter regions, followed by intragenic and intergenic regions (Figure 2B).

We analysed differential methylation of CGIs, using only those CGIs for which we obtained a methylation value in all samples. Of the 9776 analysed CGIs, 204 showed differential methylation between NAT and primary tumour samples (*n* = 5; Figure 2C, Appendix A). When assessing the genomic location of these differentially methylated CGIs, similar proportions were found as for all CGIs (Figure 2B), indicating that differentially methylated CGIs are not enriched for a specific category. The majority of differentially methylated CGIs were part of the lowly methylated CGI category and were hypermethylated in primary tumour and LNM samples compared to NAT (*n* = 3; genes = 186; Figure 2C,D). No differentially methylated CGIs were found in LNM samples, compared to NAT or primary tumours CGIs (*n* = 3). However, when assessing DNA methylation levels of the differentially methylated CGIs found in primary tumour samples, the average methylation in LNM was higher than in NAT, indicating that part of the tumour signature is still evident in the LNM samples (Figure 2E and Appendix A). The average CGIs methylation was higher in primary tumour compared to NAT. CGI methylation decreased in LNM compared to primary tumour, but remained higher than NAT (*n* = 3; Figure 2F), again suggesting a milder but retained methylation phenotype in metastasis compared to primary tumour (*p <* 0.001). To assess the consistency of the differentially methylated CGIs, methylation levels of NAT and primary tumour levels of all five individual patients were compared (*n* = 5; Figure 2E and Appendix A). Even though some degree of heterogeneity between patients was observed, especially for Patient 3, Patient 1 did show hypermethylation in most of the identified differentially methylated CGIs at the primary tumour stage. Furthermore, this hypermethylation was maintained to a higher degree in the LNM sample of Patient 1 compared to Patients 2, 4 and 5 (Figure 2E). This indicates that even though the magnitude of methylation changes can vary between patients, certain genomic regions may be more susceptible to changes than others.

As previously the majority of methylation changes in colon cancer have been observed in the regions surrounding CGIs rather than CGIs themselves [22], we also analysed differential methylation in these so-called CGI shores and shelves. The CGI shores were defined as the 2 kb regions surrounding the CGIs directly and the CGI shelves as regions 2–4 kb up- and downstream of CGIs. Of the 7144 CGI shores for which we had a methylation value in all samples, 46 showed differential methylation between NAT and primary tumour samples (Appendix A). All but two of the differentially methylated CGI shores were hypermethylated in primary tumour samples and most have low to intermediate methylation in NAT, similar to what is observed in the CGIs. The same tendency was observed for CGI shelves: Out of 5133 analysed CGI shelves, 66 were differentially methylated between NAT and primary tumour samples and of these, 65 were hypermethylated in the primary tumour (Appendix A). The number of differentially methylated CGI shores and shelves was lower compared to differentially methylated CGIs, which is different from previous reports [22]. This discrepancy may be a result of the different methods used, as earlier studies using methylation arrays will have had a different CpG coverage in these areas of the genome than methyl-seq.

As CGIs often overlap gene regulatory elements, such as promoters, and are therefore more likely to have a functional relevance, we continued our analysis with the differentially methylated CGIs. We annotated the differentially methylated CGIs to their overlapping genes or to genes within a 1.5 kb proximity of the CGI. Only CGIs with a minimal methylation difference of >20% in primary tumour (% CGI primary tumour–% CGI NAT) or in both primary tumour and LMN (% CGI LNM–% CGI NAT) compared to NAT were used for further analysis, as a smaller methylation difference is unlikely to be of biological significance for gene regulation [23,24] (Appendix A).

As CRC is a heterogeneous disease and differences in methylation phenotype, are associated with different prognoses, we evaluated the heterogeneity of the tumours in this study to discard MSI tumours based on the CIMP classification. For this, we examined if the genes closest to our differential methylated CGIs corresponded to any of the genes of the CIMP panel (*MLH1*, *CDKN2A*, *MINT1*, *MINT2*, *MINT31*, *CACNA1G*, *CRABP1*, *IGF2*, *NEUROG1*, *RUNX3*, *SOCS1*, *HIC1*, *IGFBP3* and *WRN* [25,26]. Only *RUNX3* was identified and had an average methylation difference of 60.9% between primary tumour and NAT samples (*n* = 5). Therefore, all tumours analysed were classified as CIMP-low.

Of the 195 hypermethylated CGIs hypermethylated in primary tumour compared to NAT, 60 CGIs were also hypermethylated >20% in LNM. *LMO1*, *RP11-834C11.12*, *MAN1A2P1*, *HTRA3* and *ESR1* were the top five differentially hypermethylated genes in primary tumours compared to NAT (Appendix A). Gene ontology (GO) analysis using Metascape found 152 genes enriched for 40 biological processes (Appendix A). Circus plot analysis showed that primary tumour and NAT share 106 genes (Figure 3A). This clearly demonstrates the signature of the differential methylation in primary tumour (Figure 3B). These differences are evident also in the enriched ontology clusters, where they highlight the lack of genes associated with mesenchyme development in NAT (Figure 3C).

The most methylated genes in LNM were *OPRL1*, *NUDT11*, *STAC2*, *LLfos-48D6.2* and *RP11-281J9.2* (Appendix A). Gene ontology (GO) analysis using Metascape identified 42 genes enriched in 38 biological processes (Appendix A). Circus plot analysis showed that all 42 CGIs from LNM overlapped the hypermethylated CGIs from the primary tumour samples (Figure 3D, Appendix A). In addition, primary tumour and LNM shared some gene ontology terms (Figure 3E,F and Appendix A). These results reiterate that the primary tumour signature is evident in the LNM samples, which could provide information about possible genes or markers that contribute to the spreading routes from the primary tumour to lymph nodal metastases in colorectal cancer. The most significantly enriched terms after gene ontology analysis, shared by primary tumour and metastasis were: synapse organisation, behaviour, learning and memory, synaptic signalling, neuronal system multicellular organismal response to stress and regulation of ion transport (Figure 3F, Appendix A). However, gene ontology also showed enrichment in the biological processes of mesenchyme development, tissue morphogenesis and stem cell differentiation. In addition, through a heatmap we identified that enriched genes had stable differentially methylated CGI methylation (higher, equal or slightly less, but not similar to NAT) between primary tumour and LNM (Figure 3G,H). This analysis showed that most of the enriched genes have low methylation in NAT, but high methylation levels in primary tumour and LNM.

### 2.5. Validation of Hypermethylated Metastasis CGIs in a Larger Cohort

Next, we aimed to validate the 60 hypermethylated CGIs, which are shared between primary tumour and LNM samples using The Cancer Genome Atlas (TCGA) cohort [1]. For this, we analysed methylation array data from 74 CRC patients with matched control and tumour samples. Out of the 60 CGIs hypermethylated in both primary tumour and LNM, 48 were covered by the arrays. Average methylation of all patients showed >20% methylation difference between control and tumour samples in 38 out of 48 CGIs (79.2%; adjusted *p*-value < 0.05 for all; Appendix A), showing a high overlap between the changes observed in our patient group and the larger TCGA cohort. When we looked at methylation changes in individual patients, 35 CGIs showed >20% hypermethylation differences in half of the cohort, and seven CGIs in over 85% of patients. Therefore, any of these 35 CGIs could be a potential biomarker for CRC prognosis. Based on the bimodal methylation pattern of CGIs (Appendix A), we set the cut-off for selecting candidate genes as ≥40% differences in methylation, based on the fact that the majority of CGIs overlapped promoters, in which methylation changes may affect gene expression and have a biological impact. To associate CGIs’ methylation changes to tumour prognosis and cancer progression, we analysed patient survival in the TCGA cohort. We identified five CGIs where methylation levels correlated significantly with patient survival overlapping the genes *FIGN*, *HTRA3*, *BDNF*, *HCN4* and *STAC2* (Table 1). Statistical association between poor survival and hypermethylation of CGI was established in *FIGN* (*p =* 0.030), *HTRA3* (*p =* 0.009), *BDNF* (*p =* 0.002), *HCN4* (*p =* 0.018) and *STAC2* (*p =* 0.007) and may be used as potential markers for metastasis progression in colorectal cancer (Figure 4; Table 1).

## 3. Discussion

Because CRC is influenced mainly by environmental factors and the sporadic nature of CRC results in high tumour heterogeneity, improved patient characterisation is important to optimise diagnosis, prognosis and treatment of patients. This includes molecular classifications, such as the consensus molecular subtype (CMS) for RNA expression and the CIMP high/low classification for methylation [1,27,28]. However, these classifications focus mainly on primary tumours, whereas genome-wide methylation patterns of CRC metastasis have remained uncharacterised.

In our study, we investigated the methylation landscape of CRC primary tumours and compared it to normal adjacent tissue (NAT) and lymph node metastasis (LNM), in order to characterise the DNA methylation changes occurring, throughout colon cancer progression and to identify potential biomarkers for the identification and prognosis of CRC.

As described before in many types of cancers, global methylation decreases in colon primary tumours compared with normal adjacent tissues [11,12,29]. In accordance with this, we also saw losses in global DNA methylation in primary tumour samples. Surprisingly, we observed an increase of global methylation levels in lymph node metastasis compared to primary tumour. This had previously been reported in renal cancer metastasis, but so far not in colorectal cancer [30]. In other types of cancers, non-CGI regions have been described to lose DNA methylation in primary tumour compared to NAT [31,32]. We also observed small losses in non-CGI regions of the genome of primary tumours. In previous studies, the affected areas differed [22,32,33]. In cytogenetically normal acute myeloid leukaemia hypomethylation occurred mainly in areas far from CGIs (open sea regions) [33], whereas in colon cancer methylation changes were found predominantly in the CGI shores [22]. It is possible that these differences are at least partly caused by different methylation array assays used in the different studies, as they might cover different parts of the genome. In the present study, using next generation sequencing, we covered a much larger fraction of the genome and especially of the non-CGI regions. We observed hypomethylation over large regions in areas with low CpG density, which correspond to the open sea regions. These hypomethylated regions can potentially be used as prognostic markers, as was shown in glioblastoma, where a prognostic signature was reported based on DNA hypomethylation of three CpGs in non-CGI open sea regions [31]. Analysis of epigenetics pan-cancer alterations have demonstrated a consistent pattern of methylation dysregulation across tumour types, where the highest dysregulation occurred at non-CGI regions [32]. Changes in non-CGI methylation could have functional implications for the cancer cell, as DNA methylation is an important hallmark of heterochromatin structure in normal cells and may be required for genome integrity. However, in cancer cells non-CGI hypomethylation has so far been associated with the euchromatin, which is associated with gene expression changes [31]. In fact, the most dynamic changes in gene expression occur in genes regulated by non-CGI promoters during normal cell differentiation [33]. Furthermore, in human embryonic stem cells a high methylation of non-CGI regions has been described, similarly, to our discoveries in normal adjacent tissues [34].

Our results show that LNM has higher methylation in non-CGI regions than primary tumours. There are several options to explain this unexpected finding. First, it may be that the cells represent a subpopulation of the primary tumour, such as cancer initiating cells (CICs), also known as cancer stem cells. CICs represent the cells responsible for metastasis and have the capacity of self-renewal, differentiation and migration [35]. Although CICs are essential for tumour initiation, progression, and metastasis, they represent <5% of the total number of cells within various solid tumour types, which would explain the low methylation of non-CGI regions found between LNM and primary tumours [36]. Single-cell analysis would be required to analyse methylation of CICs within primary tumour samples. Other explanations are that the cells forming the metastasis split from the primary tumour at an earlier stage, before the majority of DNA methylation changes occurred. In fact, the hypermethylation of LNM compared to primary tumour has been reported before in renal cancer metastasis [30]. This would mean that the majority of DNA methylation changes are not the driving factors of the metastatic process. Furthermore, we cannot exclude the possibility of immune cell infiltration into the LMN samples, resulting in somatic cell contamination. However, so far, there is no bioinformatic tool to deconvolute the lymphocytes methylation patterns, as exist for RNA sequencing [37]. As we are not the first study, to report that metastatic samples have smaller DNA methylation changes than primary tumour samples, it will be important to investigate the possible reasons for this pattern in future studies.

In neuroblastoma, lower methylation of non-CGI regions in have been associated with poor prognosis [38]. We also found low non-CGI methylation levels in colon cancer, which may be a consequence of the advanced CRC stage of the patients analysed, with at least one lymph node metastasis, which is associated with a poor survival. On the other hand, the high levels of non-CGI methylation in NAT could be a result of the intestine containing two stem cell populations [35]. However, further analysis will be necessary to prove these hypotheses, given most investigations in cancer have not focused on non-CGI regions and their potential as regulator of many processes during carcinogenesis, partly because early methylation arrays did not cover these areas.

As described before for cancer, we also found hypermethylation in many CGIs in primary tumours compared to NAT. In 2017, Vidal et al. found differentially methylated regions (DMR) hypermethylated of CGI regions in colon primary tumour compared to matched healthy tissue, which gained further methylation in liver metastasis [39]. Despite this increase in DNA methylation, the number of hypermethylated differentially methylated regions increased only marginally, from 12,364 to 15,373, between primary colon tumour and matched liver metastases [39]. In contrast, we saw a decrease of CGI hypermethylation in LNM compared to primary tumour. Our finding is more similar to a study performed by Ju, et al. where the number of hypermethylated genes in stages I–III CRCs were significantly larger than that of stage IV CRCs and liver metastasis [20]. It may be that the methylation profile of metastasis in lymph node is the first step in colon cancer cell migration and invasion. Studies have shown that during epithelial-mesenchymal transition (EMT) extensive genome hypomethylation occurs along with a restricted CGI hypermethylation in prostatic cancer cells [40]. These reversible CGI methylation changes regulate the transcriptional control of EMT and CICs driver genes [40]. Further analysis, such as, single cell methylation sequencing over colon CICs should reveal novel information to confirm our theory and also discard possible background data from normal cells (fibroblast, leukocytes, etc.).

Additionally, we assessed whether our differentially methylated CGIs overlapped CIMP-associated genes to classify tumours as CIMP high or CIMP low. We found only one gene (*RUNX3*) hypermethylated in the colon primary tumours of our five patients. We therefore classified the analysed tumours as CIMP low, which is associated with poor prognosis [15]. This is in line with all our tumours being metastatic, and them being histological classified as tubular adenocarcinomas [41].

The genes overlapping the most hypermethylated CGIs found in colon primary tumour compared to NAT were *LMO1*, *RP11-834C11.12*, *MAN1A2P1*, *HTRA3* and *ESR1.* The most enriched GO terms in primary tumour were related with learning and behaviour, highlighting major differences with NAT in mesenchyme development and stem cell differentiation, as described in literature [42]. Next, we sought to identify differentially methylated CGIs of primary tumours that maintain their hypermethylation in metastasis, as potential markers of CRC progression. We identified 42 CGIs which were hypermethylated (>20%) in primary tumour and LNM. The most methylated genes found in colon primary tumour compared to LNM were *OPRL1*, *NUDT11*, *STAC2*, *LLfos-48D6.2* and *RP11-281J9.2.* GO analysis showed that shared hypermethylated genes in primary tumour and LNM were mainly related to synaptic and neural activity. Interestingly, these biological processes have been previously associated with CICs [43,44]. In addition, we found enrichment of other biological processes, including mesenchymal development, tissue morphogenesis and stem cell differentiation, which are known mechanisms for EMT and metastasis [45]. These last processes are less enriched in LNM than in primary tumours. Although one of the objectives of this investigation was to find similarities in the methylation profiles of primary tumours and LNM, in order to find prognostic biomarkers, it is also important to highlight the differences between the two. The study of these differences could lead us to a better understanding of the metastatic process in colon cancer.

In order to validate our putative CGI biomarkers in a larger cohort, we included methylation array data from 74 further CRC patients from the TCGA cohort [1] and linked the DNA methylation changes to disease prognosis. The majority of the analysed CGIs also showed hypermethylation in the TCGA cohort, confirming them as potential biomarkers for CRC. Furthermore, survival analysis showed that CGI hypermethylation of the genes *FIGN*, *HTRA3, BDNF, HCN4* and *STAC2* is related to poor patient survival, making them potential markers for progression of CRC. Two of these genes, *HTRA3* and *STAC2*, were found as the top five hypermethylated CGIs in primary tumour and LNM, respectively. *FIGN* (fidgetin) is a gene coding for a microtubule severing enzyme as well as a depolymerase, predominantly involved in mitosis but also contributing to cell migration and neuronal development [46]. Studies have shown that *FIGN* is overexpressed in human hepatocellular carcinomas promoting hepatocyte invasion [47] *HTRA3* (high temperature requirement A3) is a member of the HtrA family and is a proapoptotic protease shown to promote drug-induced cytotoxicity and proposed to have an antitumour effect [48]. Furthermore, HtrA3 is an inhibitor of the bone morphogenetic protein pathway, linked to EMT [49], and has been implicated as a tumour suppressor gene during cancer progression in many cancers, such as lung cancer, ovarian cancer, breast cancer and colorectal cancer [49,50,51,52]. Specifically, in colon cancer, low immunohistochemical expression of HtrA3 has been associated with invasion and poor prognosis [49]. The low expression of HtrA3 in colon cancer tissues could be linked to a repression by methylation, which would be interesting for further investigations. *BDNF* (brain-derived neurotrophic factor) is a well-studied growth factor that serves many critical functions within the central nervous system and has important functions regarding development, morphology and synaptic plasticity in the brain [53]. In triple-negative breast cancer, the interaction of estradiol with *BDNF* and its receptor, promotes brain metastasis [54]. Interestingly, we found that *ESR1,* encoding oestrogen receptor 1, was also highly methylated in primary tumour and LNM. In cervical cancer, *BDNF*/TrkB signalling is overexpressed, and associated with poor prognosis, invasion, migration and EMT [55]. *BDNF* one of the most interesting genes for further studies, especially regarding its biological implications for colon cancer migration and CICs. GO showed that *BDNF* is associated to mesenchymal development. Mesenchymal development is less enriched in LNM than in primary tumour. The *BDNF* gene has been reported to be highly expressed in bone marrow mesenchymal stem cells [56], again supporting a potential function in CICs’ behaviour of metastasis. It will be important to evaluate how methylation can affect the function of *BDNF* in the colon metastatic process. *HCN4* (hyperpolarisation activated cyclic nucleotide gated potassium channel 4) has been reported to contribute significantly to the generation of basic cardiac electrical activity in the sinus node and is a mediator of modulation by β–adrenergic stimulation [57]. As a potassium channel its hypermethylation could affect ion transport processes, which were enriched in GO analysis. *STAC2* (Src homology 3 and cysteine-rich domain 2) is a gene member of the *STAC* gene family, encoding a protein containing an SH3 domain and a zinc finger domain, which is usually expressed in the central nervous system and has been associated with calcium channel regulation [58,59]. To date, two investigations have shown a downregulation of this gene in breast cancer [60,61]. *STAC2* may be an interesting biomarker, as it has a known biological function in cancer progression. *HCN4* and *STAC2* have otherwise not been reported in cancer research so far, especially not regarding their methylation status. In contrast, *FIGN*, *HTRA3* and *BDNF* have been reported in cancer, EMT, invasion/migration process or poor outcome of patients, but our study for the first time links them to methylation status. Here we demonstrated that CGI hypermethylation of *FIGN*, *HTRA3, BDNF, HCN4* and *STAC2* in colorectal cancer is associated with patient progression to metastasis, identifying them as putative future biomarkers for CRC progression.

## 4. Materials and Methods

### 4.1. Discovery Cohort

For whole-genome analysis of DNA methylation, five patients diagnosed with sporadic CRC and at least one lymph node metastasis (tumour stage between IIIB–IIIC, according AJCC Cancer Staging Manual, Eighth Edition) were recruited [62]. The participants had not received treatment and were invited to participate only because the tumour was to be removed by medical recommendations. All participants signed an informed consent approved by the ethics committee of the Universidad de La Frontera, Temuco, Chile, prior to sample collection. Formalin-fixed paraffin-embedded (FFPE) tissue specimens of CRC, normal adjacent tissue (NAT) and at least one lymph node metastasis (LNM) tissue samples were obtained by surgery. If there was no macroscopic evidence of metastasis in the lymph node, the specimen was selected after fixation and histopathological analysis at the Unit of Pathological Anatomy and Cytology of Hospital Hernán Henríquez Aravena, Temuco, Chile. The diagnosis was confirmed by histological examination (biopsy) performed by a pathologist from the same Unit. DNA extraction and procedures were performed in the Laboratory of Integrative Biology (LIBi), Universidad de La Frontera, Temuco-Chile.

### 4.2. DNA Extraction

The extraction of genomic DNA was performed following manual microdissection of ten tissue sections for each sample (5 NAT, 5 primary tumours and 5 LNM). Prior to sequencing, samples were assessed by an expert pathologist. A section of the tissues was stained with Haematoxylin-Eosin (Appendix A) to mark the area with the tumour (primary tumour or lymph node metastasis) or to verify the normal tissue. Then samples were microdissected specifically in the selected area for DNA extraction. Unstained 5 μm-thick sections were cut from each FFPE tissue, de-waxed with Histo-Clear (National Diagnostics, Atlanta, GA, USA) and re-hydrated through graded concentrations of ethanol. After dissection, DNA extraction was performed using the QIAamp DNA FFPE tissue kit (Qiagen, Germantown, MD, USA), following manufacturer’s instructions. DNA pellets were resuspended in Tris-EDTA Buffer and stored at −20 °C until further use.

### 4.3. DNA Quality Control Prior Sequencing

The quality of extracted DNA was measured on the Infinite 200 PRO NanoQuant (Tecan, Männedorf, Switzerland) and by electrophoresis. DNA concentrations were over 20 ng/µL and had an OD ratio A260/A280 ≥ 1.8 and A260/A230 ≥ 1.9. The integrity of DNA was assessed by amplification of a 268-bp fragment of the β-globin gene using GH20 and PCO4 primers.

### 4.4. Genome Wide Methyl-Seq Bisulphite Sequencing

To determine and compare the methylation status across the genome from primary CRC tumour and corresponding NAT and LNM, 500 ng of extracted DNA were analysed by Enhanced Methyl-Seq (Epigentek, Farmingdale, NY, USA). This platform is a bisulphite sequencing method to detect 7–8 million unique CpG sites, covering nearly all CpG islands, gene promoters and most genetic regulatory elements, gene bodies, and repeated DNA sequences. The workflow of samples for the sequencing was as follows: DNA was quantified by fluorescence. For each sample 300 ng of DNA was digested for 2 h with MSP1 enzyme (20 U/sample) at 37 °C followed by 2 h with Taqal (20 U/sample) at 65 °C. The digested DNA was size selected with <300 bp DNA fragments (CGI enriched) that were collected for bisulphite treatment. DNA fragments were bisulphite treated with the Methylamp DNA Bisulphite Conversion Kit (Epigentek, Farmingdale, NY, USA). Conversion efficiency of bisulphite treated DNA was determined by real time PCR using two primer pairs with using control DNA. The first primer pair binds to bisulphite-converted DNA (β-actin) and the second primer pair to unconverted DNA (GAPDH) on the same bisulphite-treated DNA sample. DNA was >98% converted. Bisulphite treated DNA was ligated with adaptors using a random probing method. Then, libraries were amplified using indexed primers, followed by library purification. Purified library DNA was eluted in 12 μL of water. Library quality and quantity was analysed on a Bioanalyzer and by KAPA qPCR Library Quantification. Libraries (15 nM per sample) were sequenced by next generation sequencing on an Illumina HiSeq 4000 (Thermofisher, Waltham, MA, USA).

### 4.5. Library Mapping and Trimming

Raw fastq sequence files were quality trimmed and then adaptor trimmed with Trim Galore v0.4.2 using the “--clip_r1 9” function. Mapping and methylation calling of bisulphite-sequencing data was carried out using Bismark v0.19.1 against the human GRCh38 genome assembly with the --pbat mode. Sequencing data were deposited under accession code GSE151318 in the Gene Expression Omnibus database [63].

### 4.6. DNA Methylation Sequencing Analysis

DNA methylation was quantified and analysed in SeqMonk (https://www.bioinformatics.babraham.ac.uk/projects/seqmonk/), using a tile-based method binning consecutive genomic windows with a fixed CpG number (200 CpG windows). The fixed CpG windows ensure similar data coverage of windows independent of genomic CpG density. Methylation values were quantified with the bisulphite-sequencing pipeline quantification, which calculates per-base methylation percentages and then averages these within each window. Windows were filtered to ensure a minimum coverage of 10 observed cytosines per probe window. Only windows with the required minimal coverage in all samples were taken into account, allowing for the assessment of 89.4% of all 200 CpG windows (*n* = 128,824). Non-CGI regions were determined by filtering for 200 CpG windows not overlapping CGIs (*n* = 112,695). CGIs were defined using the CpG island features from Ensembl v90. Hypomethylated regions were defined as 200 CpG windows with an average methylation difference between NAT and primary tumour samples of >10%. Sequence composition of hypomethylated regions was assessed by Compter (https://www.bioinformatics.babraham.ac.uk/projects/compter/), which analysed enrichment or depletion of k-mers in the extracted sequences in hypomethylated and in the non-changing regions. To build the heatmaps and perform the clustering, the --subset option was specified, instructing Compter to extract 1000 sequences distributed at regular intervals throughout the input FASTA files.

CGI methylation was assessed by directly quantifying methylation of CGIs, using the same bisulphite-sequencing pipeline with the same filters for coverage as used for 200 CpG windows. In total, 46% of all CGIs had sufficient coverage for analysis when including all samples of Patients 2, 4 and 5 (*n* = 9776). When comparing NAT and primary tumour samples of all patients (Patient 1 to 5), 31.1% CGIs were included for analysis (*n* = 7009). CGIs were split into lowly methylated (<40%) and highly methylated (>80%) CGIs, depending on their methylation level distribution in normal samples (Appendix A). Differentially methylated CGIs were identified using the EdgeR tool in SeqMonk, using a *p*-value cut-off of <0.05 and applying multiple testing corrections with Benjamini and Hochberg False Discovery Rate correction. Genomic localisation of CGIs (promoters, intragenic, intergenic) was defined using the gene features from Ensembl v90. Promoters were identified to include the region 1.5 kb upstream and the first 500 bp within the gene. CGIs overlapping this region were defined as promoter CGIs. This annotation only captures the upstream gene promoter and does not consider intragenic, alternative promoters. Intragenic CGIs were set to be CGIs overlapping genes, but excluding previously identified promoter CGIs. Intergenic CGIs were defined as not overlapping genes and not overlapping promoters.

Differentially methylated CGIs were filtered for an average methylation difference of >20% between NAT and primary tumour samples, in order to identify changes with possible gene regulatory function. As no differentially methylated CGIs between primary tumour and LNM samples were identified, it was instead tested which primary tumour differentially methylated CGIs maintained their differential methylation in LNM samples. For this, differentially methylated CGIs were filtered for those with >20% methylation difference between NAT and primary tumour and between NAT and LNM samples. CGIs were annotated to overlapping genes or genes within a 1.5 kb proximity.

### 4.7. Gene Ontology Analysis

For gene ontology (GO) analysis and signalling pathway analysis, gene enrichment analysis of genes annotated to differentially methylated CGIs (differential methylation over 20% between NAT/primary tumour and NAT/LNM) was performed, using the Metascape web-based portal (http://metascape.org/gp/index.html#/main/step1) [64], uploading an Excel file containing two list of candidate CGIs (primary tumour and or primary tumour and LNM).

### 4.8. TCGA Methylation Array Data

Publicly available methylation array data, part of the The Cancer Genome Atlas Colon Adenocarcinoma (TCGA-COAD) project1, were downloaded from the Genomic Data Commons (GDC) Data Portal (https://portal.gdc.cancer.gov). Only patients with both normal tissue and primary tumour samples were considered for analysis. Data were available on two different Illumina platforms, the HumanMethylation27 BeadChip (27k array) and the HumanMethylation450 BeadChip (450k array). Methylation was quantified by averaging the beta values of the individual CpGs overlapping the CGI. These values were used to calculate the difference in methylation between the primary tumour and the normal sample. The 27k array covered 18 out of our 60 candidate CGIs and the 450k array covered 41 CGIs, enabling methylation analysis of 48 candidate CGIs.

### 4.9. Candidates Genes Selection and Survival Analysis

In order to identify biomarkers associated with prognosis, we used the data from the methylation arrays of 74 CRC patients with matched control and tumour samples available in the TCGA database. Subsequently, we defined CGIs with low and high methylation using a cut-off point of 40% based on the bimodal methylation pattern of CGIs (Appendix A). Then, Kaplan–Meier survival curves were performed for cases with low versus high methylation of CGIs. CRC cases with an overall survival date shorter than 30 days were omitted from the study (*n* = 10). For the survival analysis, “survival” was defined as the time between the first surgery and the end of follow-up or death due to CRC.

### 4.10. Statistical Analysis

Statistical analysis was conducted using GraphPad Prism 8 (San Diego, CA, USA). Mean methylation of non-CGI regions between NAT and tumour samples was compared using one-way ANOVA followed by a Tukey’s multiple comparison test. The difference in methylation of the 48 candidate CGIs in the TCGA cohort was tested using a *t*-test followed by a Bonferroni multiple comparison correction. Kruskal–Wallis followed by Dunn’s test was performed to compare NAT, primary tumour and LNM methylation levels of CGIs. For Kaplan–Meier survival curves were constructed using the statistical package SPSS version 22.0 (SPSS Inc., Chicago, IL, USA). The difference between the survival curves were analysed using the log-rank test. A *p*-value < 0.05 was considered statistically significant.

## 5. Conclusions

In summary, our analysis found global hypomethylation and CGIs hypermethylation of colon primary tumour compared to normal tissues; which is in line with our general understanding of the cancer methylome. However, for the first time the methylation changes in lymph node metastasis from colorectal cancer are described. We described a methylation profile similar to cancer-initiating cells (CICs) in lymph node metastasis; which could promote epithelial-mesenchymal transition, invasion and migration. Although further investigations will be required to elucidate the full mechanism; our results suggest a new methylation mechanism that could help explaining the metastatic process.

Even though some heterogeneity between at the CGI methylation changes of individual patients, a subgroup of CGIs showed very consistent changes, which were confirmed in a larger cohort of patients. These CGIs have the potential to become biomarkers used for cancer screening. Furthermore, for five genes (*FIGN*, *HTRA3*, *BDNF*, *HCN4* and *STAC2*) CGI hypermethylation was associated with poor patient survival, making them putative prognostic markers for CRC.

## Figures and Tables

**Figure 1 cancers-12-02710-f001:**
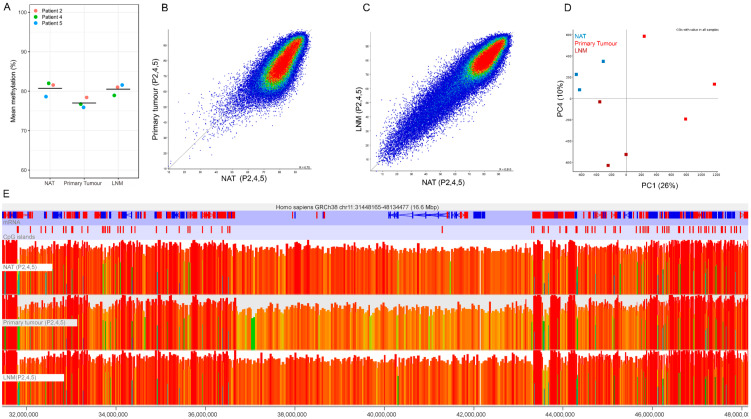
Methylation patterns of non-CGI regions in normal colon tissue (NAT), primary tumour and lymph node metastasis (LNM) samples of three CRC patients. (**A**) Mean methylation of non-CGI regions for each individual patient. The black line indicates the mean of the three patients (one-way ANOVA, *p =* 0.1360). (**B**) Scatterplot comparing methylation levels between NAT and primary tumour samples and (**C**) between NAT and LMN samples. Each dot indicates the average methylation of a 200 CpG window (excluding CGIs). (**D**) Principal component analysis showing clustering of normal, primary tumour and LNM samples on PC1 and PC4. (**E**) Screenshot of the genome showing large regions with lower methylation levels in primary tumour samples compared to NAT and LMN samples. Each bar represents a 200 CpG window. The height and colour of the bar indicate the methylation values. NAT: normal adjacent tissue; LNM: lymph node metastasis.

**Figure 2 cancers-12-02710-f002:**
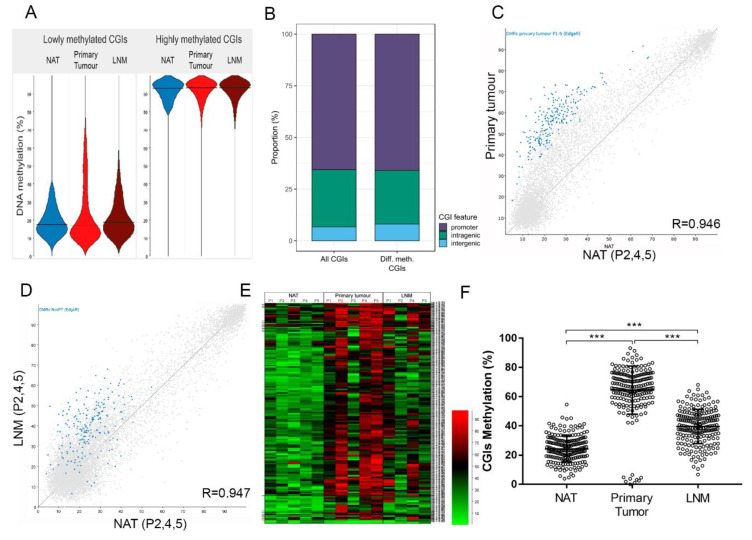
CGI methylation levels in normal adjacent tissue, primary tumour and lymph node metastasis samples of CRC patients. (**A**) Beanplots showing DNA methylation levels of lowly (<40%) and highly (>80%) methylated CGIs (*n* = 9776). (**B**) Bar chart showing the proportion in which different CGI features are represented in all analysed CGIs (*n* = 7009) and in differentially methylated CGIs (*n* = 246). (**C**) Scatterplot comparing DNA methylation levels of CGIs between NAT and primary tumour samples and (**D**) between NAT and LMN samples in which each dot represents a CGI (*n* = 7009). Differentially methylated CGIs are highlighted in blue. (**E**) Heatmap showing methylation levels of 246 differentially methylated CGIs for each individual patient. Each bar represents a CGI and the colour scale indicates its methylation level. (**F**) Strip chart comparing methylation percentage of 204 CGIs in NAT, primary tumour and LNM samples (1.5 kb upstream to coding-genes) (*** *p* < 0.001, Kruskal–Wallis followed by Dunn’s test). NAT: normal adjacent tissue; LNM: lymph node metastasis.

**Figure 3 cancers-12-02710-f003:**
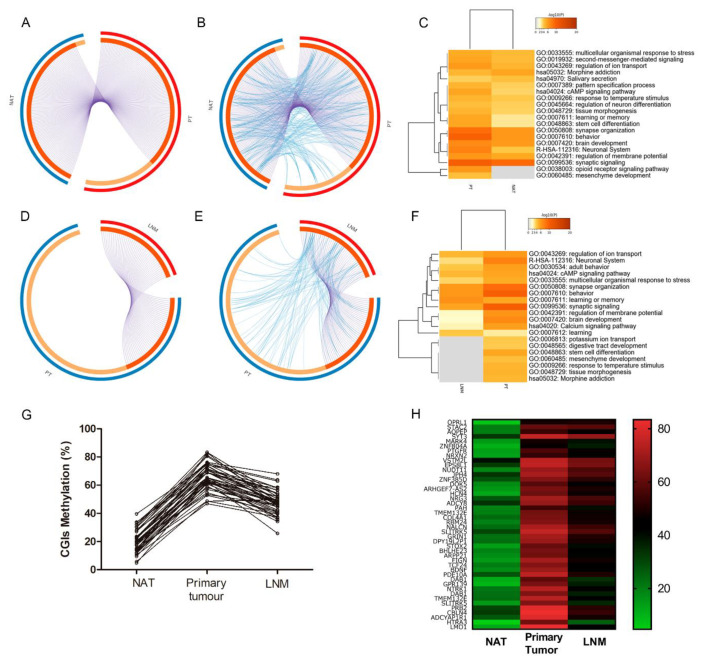
Gene ontology analysis comparing hypermethylated CGIs between NAT, primary tumour and LNM. (**A**) The circus plot shows how genes from the input gene lists overlap genes with differentially methylated CGIs enriched for ontology terms of primary tumour and NAT. On the outside, each arc represents the identity of each gene list. On the inside, each arc represents a gene list, where each gene has a spot on the arc. Dark orange colour represents the genes that appear in multiple lists and light orange colour represents genes that are unique to that gene list. Purple lines link the same genes that are shared by multiple gene lists. (**B**) Circus plot showing the shared term level, where blue curves link genes that belong to the same enriched ontology term, between primary and NAT. The inner circle represents gene lists, where hits are arranged along the arc. Genes that hit multiple lists are coloured in dark orange, and genes unique to a list are shown in light orange. (**C**) Dendrogram showing enriched ontology clusters across primary tumour and NAT. The colour scale of the heatmap cells represents their *p*-values, white cells indicate the lack of enrichment for that term in the corresponding gene list. (**D**) The circus plot shows how genes from the input gene lists overlap genes with differentially methylated CGIs enriched for ontology terms (42 genes) of primary tumour and LNM. (**E**) Circus plot showing the shared term level, where blue curves link genes that belong to the same enriched ontology term between primary tumour and LNM. (**F**) Dendrogram showing enriched ontology clusters across input gene lists, between primary tumour and LNM. (**G**) Line chart showing the differentially methylated CGIs across the colon tissues (NAT, primary tumour and LNM). (**H**) Heatmap representing the methylation percentage of 42 CGIs in NAT, primary tumour and LNM. (NAT: normal adjacent tissue; PT: primary tumour; LNM: lymph node metastasis).

**Figure 4 cancers-12-02710-f004:**
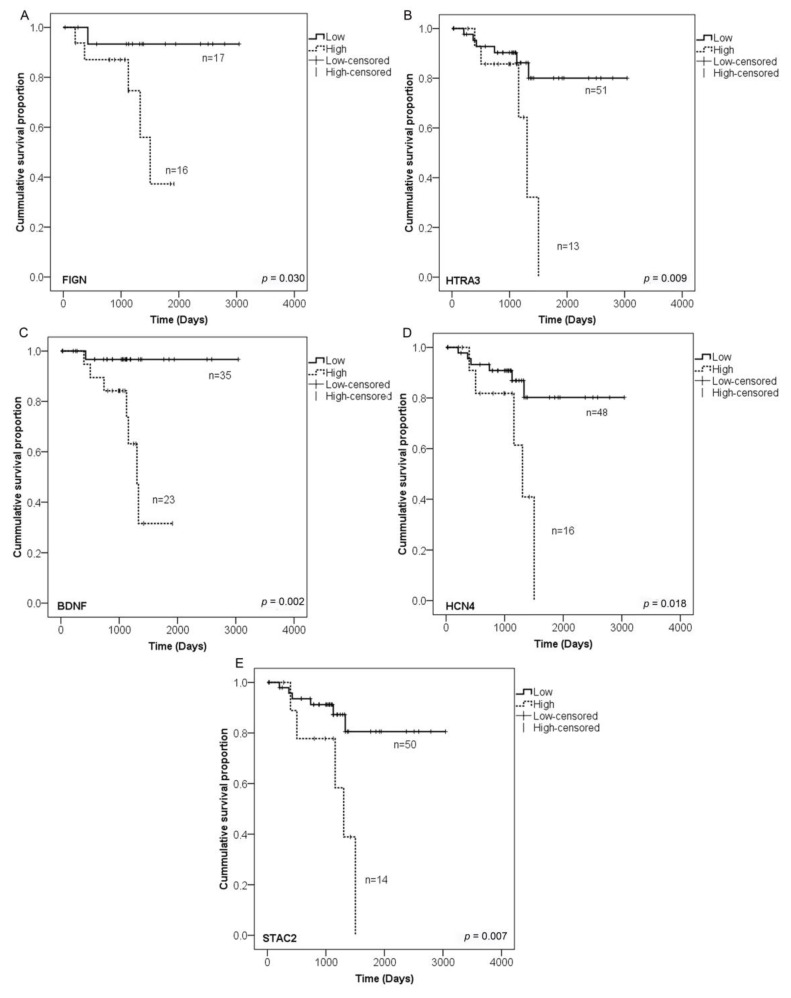
Kaplan–Meier survival curves for patients with colorectal cancer from The Cancer Genome Atlas (TCGA) cohort. The solid lines indicate patients with low levels of CGI methylation and the dotted lines indicate patients with hypermethylation of CGIs overlapping candidate genes. (**A**) *FIGN* (*p =* 0.030); (**B**) *HTRA3* (*p =* 0.009); (**C**) *BDNF* (*p =* 0.002); (**D**) *HCN4* (*p =* 0.016) and (**E**) *STAC2* (*p =* 0.007). Stratified log-rank test.

**Table 1 cancers-12-02710-t001:** Methylated candidate CGIs for colorectal cancer progression.

Nearest Gene	Description	Chr	Start	End	Feature Strand	CGI Orientation in Relation to Nearest Gene	Proposed Function
*FIGN*	Fidgetin	chr2	163,736,135	163,737,001	−	downstream	ATP-dependent microtubule severing protein
*HTRA3*	HtrA serine peptidase 3	chr4	8,269,488	8,270,364	+	overlapping	Serine protease that cleaves beta-casein/CSN2 as well as several extracellular matrix (ECM) proteoglycans.
*BDNF*	Brain-derived neurotrophic factor	chr11	27,721,926	27,723,017	−	overlapping	Important signalling molecule that activates signalling cascades downstream of *NTRK2*
*HCN4*	Hyperpolarisation activated cyclic nucleotide-gated potassium channel 4	chr15	73,367,520	73,369,674	−	overlapping	Activated by cAMP. cAMP binding causes a conformation change that leads to the assembly of an active tetramer and channel opening.
*STAC2*	SH3 and cysteine rich domain 2	chr17	39,224,701	39,226,110	−	overlapping	Plays a redundant role in promoting the expression of calcium channel *CACNA1S* at the cell membrane, and thereby contributes to increased channel activity.

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
