# Peer review of "Landscape of Genome-Wide DNA Methylation of Colorectal Cancer Metastasis"

_cancers, 2020, doi:10.3390/cancers12092710_

Round 1

Reviewer 1 Report

In their study entitled "Landscape of genome-wide DNA methylation of colorectal cancer metastasis", Ili and colleagues assessed genome wide DNA methylation in five patients for three types of samples: primary tumor, lymph node metastasis (LNM) and normal adjacent tissue (NAT).

The manuscript is well written, though too little amount of data is present to make the story really sound.

The major issue is the lack of data. Only five patients were enrolled with three samples each, for a total of 15 samples. However two patients had demonstrated bad quality data for at least one sample type and were excluded from the main analysis, leading to only three patients (9 samples).

In this condition, it is difficult to consider any of the results reliable from a statistical point of view. In fact the difference observed between primary tumor and LNM might just be "sampling" effect instead of biological.

Beside this aspect I have several other concerns:

In paragraph 2.2, the authors identified Patient LNM as an outlier according to the PCA (Fig S2A). In the same figure it seems that one primary tumor might also be an outlier (bottom of the graph).

In paragraph 2.3, the authors mentioned they performed their analyses in CGI and non CGI regions. Which definition of CGI did the authors use?

From my understanding their non CGI analysis will contain all CpG shores and shelves which are possibly of high interest in the regulation of genes (as mentioned by the authors in the discussion). The authors should consider including these areas in the CGI.

TCGA samples used need to be identified in a supplementary table, moreover not all samples will fit for validation as they are usually primary tumors non necessary from metastatic cancers.

Other cohorts would better fit the purpose ot the authors such as

  • Luo Y, Wong CJ, Kaz AM, Dzieciatkowski S et al. Differences in DNA methylation signatures reveal multiple pathways of progression from adenoma to colorectal cancer. Gastroenterology 2014 Aug;147(2):418-29.e8.

Discussion should be more focus on the results and less on the litterature.

Several typos are also present, as for examples:

Line 133 which fewer --> with fewer

Line 275 "which define the" should be removed from the sentence 

Reviewer 2 Report

This is a very interesting study to investigate the DNA methylation signature in colorectal cancer.  The authors use paired normal adjacent tissue (NAT), primary tumor, and near tumor metastatic lymph node (LMN) tissue to compare the methylation signatures of these three types of tissues. It was noticed that the metastasis has high global and non-CGI region methylation but lower CGI methylation compared to the primary tumor. There were 5 genes associated with poor prognosis identified after the complementary analysis with TCGA cohort.

There are some comments would like the authors to address:

  1. The first and most concerning question is the percentage of tumor cell in the tested specimen. The authors used lymph nodes with metastatic cancer. However, in the methods, they did not mention if the lymph nodes were checked under the microscope and if microdissection was done to make sure only the metastatic component was used for further test. Since the study consistent showed the LMN result is between the NAT and primary tumor, it is possible due to the purity of the tissue that was used. As the authors mentioned, single cell analysis could be a better way for this comparison.
  2. This study used basically 3 LMN samples, case 1 and 3 were not included due to different reasons. If we look at the table S1, case 2, 4, and 5 are slightly different. As we now know, right sided colon cancer behaves differently compared with left sided colon cancer. The molecular signatures are different. Age could also be an important factor. Case 1 patient is 95 which means his colon cancer could have a completely different mechanism.  The authors should get mutation panel tests for these case to make sure the gene backgrounds are similar, at least to make sure there is no MSI-H or other driver mutations that may lead to biased analysise authors may want to make some changes in the article and proofread the paper again. For example, page 2 line 63, should add (CIMP), the last paragraph of introduction, figure2 page 5 line 200 AT should be NAT. The figure 2E should showed the all analyzed CGIs and differentially  methylated CGIs, but one cannot see these two CGIs in the figure  

Reviewer 3 Report

This paper appeared to be novel in its use of genome-wide methyl-seq bisulfite sequencing to investigate DNA methylation alterations associated with colorectal cancer lymph node metastasis (relative to normal colon tissue in the same patients). However, as the authors mentioned, the findings for the metastatic samples may have been influenced by infiltrating immune cells from the lymph nodes and they were unable to separate this out in their study, which reduced the significance of the results. The conclusion in the abstract and the overall study conclusion in the Discussion should be revised/tempered accordingly to account for this limitation. The sample size was also small (n=5 patients, and two of the patients were excluded from some of the analyses) and it was unclear if all of the differences highlighted in the paper were statistically significant and what approach was taken to adjust for multiple comparisons (and when). Another aspect of the paper was following up on the 60 hypermethylated CpG islands shared between the primary tumor and lymph node metastasis samples (relative to normal colon tissue) to investigate the association with survival using TCGA data, but it would be more interesting to look at CpG islands that showed different associations for primary tumor and lymph node metastasis, which the authors alluded to on p. 10 (lines 352-356). Another concern with the TCGA analysis was the exclusion of patients with an overall survival<30 days; it is unclear why this was done and would have resulted in preferentially excluding the sicker patients. More information on these and other concerns are described below.

  1. A principal limitation is that the findings for the lymph node metastasis samples may have been influenced by infiltrating immune cells from the lymph nodes and the authors were unable to separate this out in their study. The conclusion in the abstract (that “lymph node metastasis from colorectal cancer seem to have a methylation profile similar to cancer-initiating cells”) – and, similarly, the overall study conclusion on p. 14, lines 538-540 – is overstated given this possible alternate explanation and should be tempered/revised accordingly.
  2. It was unclear which (if any) of the differences highlighted in the paper were statistically significant. For ex, re Figure 1A, was the difference between primary tumor and normal tissue significant at the 0.05 level? What about the difference between lymph node metastasis and normal tissue (or between metastatic tissue and primary tumor tissue)? The same question goes for results shown in Figure 2F, for example. The significance of all findings should be clarified throughout the Results and some p-values should be presented to go along with key findings.
  3. The authors mention adjustment for multiple comparisons re differentially methylated CpG islands on p. 13, lines 486-487. What approach was used specifically, and what about adjustment for multiple comparisons in other analyses, for ex, the TCGA analyses?
  4. The authors focused on the 60 hypermethylated CpG islands shared between the primary tumor and lymph node metastasis samples (relative to normal colon tissue) to investigate the association with survival using TCGA data. However, it would be more interesting to look at CpG islands that showed different associations for primary tumor and lymph node metastasis, which the authors allude to on p. 10 (lines 352-356). The authors should focus on the CpG islands showing different associations  instead of (or perhaps in addition to) what is already presented.
  5. In the TCGA analysis, why were patients excluded who had an overall survival<30 days?
  6. The interpretation of the result for lymph node metastasis samples vs. normal tissue is a bit long and not entirely consistent. For ex, on p. 3, lines 127-128, the authors state “Surprisingly, this decrease in methylation was not observed in LNM samples, which showed similar methylation levels to NAT samples.” Then on p. 4, lines 149-151, the authors state “When analyzing the methylation of the windows driving PC4, a decrease in methylation in both primary tumour and LNM samples was observed (Figure S5). However, the decrease was more pronounced in the primary tumour than in the LNM samples.” The message is confusing – is there a decrease in LNM samples compared to normal samples or not? Again, it would help to see p-values to know if there is truly a difference.
  7. It appears that some of the analyses involved only 3 patients and others all 5. Please clarify the numbers for each analysis in the Methods/Results and why there was a difference.
  8. Some parts of the Results section include discussion, which should be moved to the Discussion section, for ex, p. 3, lines 131-136.
  9. The third paragraph of the Discussion (p. 9) is fairly long and appears to have some repetition. Suggest cutting it back to streamline the paper. Also (in same paragraph), lines 280-281, the the authors write: “As global DNA methylation is mainly comprised of non-CGIs regions, we found similar changes in both.” What is meant by “both” – i.e. both CGI and non-CGI regions?
  10. There are a few lines with awkward wording throughout the paper, for ex, p. 4, lines 154-156, and p. 4, lines 159-160.

Round 2

Reviewer 1 Report

While I agree with the authors that lymph node metastases are precious and rare and difficult to obtain for research purposes, the number of cases in the study remains very low (N=3).

Despite this issue, the manuscript can be of interest to a specific audience considering that the limitation of the low statistical power was highlighted.

The authors responded to all reviewers' comments.

Reviewer 2 Report

This is an interesting study to investigate the DNA methylation signature in colorectal cancer. The authors use paired normal adjacent tissue (NAT), primary tumor, and near tumor metastatic lymph node (LMN) tissue to compare the methylation signatures of these three types of tissues. It was noticed that the metastasis has high global and non-CGI region methylation but lower CGI methylation compared to the primary tumor. There were 5 genes associated with poor prognosis identified after the complementary analysis with TCGA cohort. 

The authors addressed the comments and concerns although there are still limits due to the sample size (only 5 cases and some analysis was done with only 3 cases which is the main concern), methods (purity of the samples, contamination of other cells in the lymph nodes although the authors now reported microdissection was done). The statistic analysis were added to help the readers to better understand the paper.